# WISER Survivor Trial: Combined Effect of Exercise and Weight Loss Interventions on Insulin and Insulin Resistance in Breast Cancer Survivors

**DOI:** 10.3390/nu13093108

**Published:** 2021-09-04

**Authors:** Nicholas J. D’Alonzo, Lin Qiu, Dorothy D. Sears, Vernon Chinchilli, Justin C. Brown, David B. Sarwer, Kathryn H. Schmitz, Kathleen M. Sturgeon

**Affiliations:** 1College of Medicine, Pennsylvania State University, Hershey, PA 17033, USA; ndalonzo@pennstatehealth.psu.edu; 2Department of Public Health Sciences, College of Medicine, Pennsylvania State University, Hershey, PA 17033, USA; lqiu@pennstatehealth.psu.edu (L.Q.); vchinchilli@pennstatehealth.psu.edu (V.C.); kschmitz@phs.psu.edu (K.H.S.); 3College of Health Solutions, Arizona State University, Phoenix, AZ 85004, USA; dorothy.sears@asu.edu; 4Department of Medicine, UC San Diego, La Jolla, CA 92093, USA; 5Department of Family Medicine, UC San Diego, La Jolla, CA 92093, USA; 6Moores Cancer Center, UC San Diego, La Jolla, CA 92037, USA; 7Cancer Metabolism Program, Pennington Biomedical Research Center, Louisiana State University, Baton Rouge, LA 70808, USA; justin.brown@pbrc.edu; 8Center for Obesity Research and Education, Temple University, Philadelphia, PA 19122, USA; dsarwer@temple.edu

**Keywords:** breast neoplasms, neoplasm recurrence, weight reduction program, resistance training, overweight, adiposity, caloric restriction, biomarkers

## Abstract

Obesity-associated breast cancer recurrence is mechanistically linked with elevated insulin levels and insulin resistance. Exercise and weight loss are associated with decreased breast cancer recurrence, which may be mediated through reduced insulin levels and improved insulin sensitivity. This is a secondary analysis of the WISER Survivor clinical trial examining the relative effect of exercise, weight loss and combined exercise and weight loss interventions on insulin and insulin resistance. The weight loss and combined intervention groups showed significant reductions in levels of: insulin, C-peptide, homeostatic model assessment 2 (HOMA2) insulin resistance (IR), and HOMA2 beta-cell function (β) compared to the control group. Independent of intervention group, weight loss of ≥10% was associated with decreased levels of insulin, C-peptide, and HOMA2-IR compared to 0–5% weight loss. Further, the combination of exercise and weight loss was particularly important for breast cancer survivors with clinically abnormal levels of C-peptide.

## 1. Introduction

Breast cancer mortality has significantly declined over the last two decades [1], leading to a growing population of breast cancer survivors. Breast cancer survivors comprise more than 50% of the 8.8 million female cancer survivors in the United States [2]. Unfortunately, the risk of breast cancer recurrence is 10–52% depending on tumor subtype and cancer stage [3]. As a result of the growing population of breast cancer survivors there has been an increase in focus on preventing breast cancer recurrence [4].

Obesity and physical inactivity are risk factors for breast cancer and breast cancer recurrence [5,6,7] with a 12% increase in risk of breast cancer diagnosis for every 5 unit increase in BMI [8]. At breast cancer diagnosis, 54–71% of women are overweight (BMI 25–30 kg/m^2^) or have obesity (BMI > 30 kg/m^2^) [9,10]. Further, weight gain after diagnosis is common, with the majority gaining weight during treatment [11]. Every five pounds gained after diagnosis is associated with a 12% and 13% increase of breast cancer-specific mortality and all-cause mortality, respectively [12]. One proposed mechanism linking obesity and breast cancer recurrence are elevated insulin levels and reduced insulin sensitivity [13,14]. Insulin resistance is associated with abdominal obesity, breast cancer specific and overall mortality [15,16], and increased breast cancer recurrence [17,18].

Exercise and weight loss reduces cancer and all-cause mortality in breast cancer patients [19,20,21,22]. It is currently unclear if exercise and weight loss interventions reliably improve metabolic markers and reduce insulin resistance in the breast cancer survivor population [23,24,25,26,27]. The WISER Survivor trial compared exercise training, weight loss and a combination of the two energetic interventions to a control group among breast cancer survivors with overweight or obesity [28,29]. In this secondary analysis, we have analyzed mechanistically relevant biomarkers. We hypothesized that the combined intervention group would lead to the greatest decrease, compared to control, in adiposity and concomitant beneficial changes in insulin, C-peptide, and insulin resistance.

## 2. Materials and Methods

### 2.1. Design

The WISER Survivor randomized controlled trial consisted of four intervention groups comparing the individual and combined effects of exercise and weight loss in breast cancer survivors with excess body weight and lymphedema. The complete study design, methods, and primary results have been published separately [28,29]. The 12-month intervention groups included: control (referred to American Cancer Society and/or their physician), exercise (weight training and aerobic exercise), weight loss (caloric restriction), and combined exercise and caloric restriction to promote weight loss.

### 2.2. Participants

Recruitment was conducted using local hospital and state tumor registries in the Philadelphia, PA, metropolitan area. All participants were breast cancer survivors with a BMI ≥ 25 kg/m^2^ but <50 kg/m^2^, younger than 80 years old, cancer free and having completed curative treatment more than 6 months before randomization, had breast cancer-related lymphedema, and sedentary lifestyle (assessed by self-report) prior to enrollment. Eligible participants had to be able to walk unaided for greater than 6 min. Exclusions included taking weight loss medication at the time of enrollment, weight loss greater than 4.5 kg in the previous 3 months, current engagement in moderate intensity exercise (e.g., bicycling or brisk walking) 3 or more times per week, weight training in the past year, and bariatric surgery. Additional recruitment methods and eligibility criteria have been published separately [30]. There were 351 women enrolled between 5 December 2011 and 21 April 2015 and all follow-up testing was performed by 28 May 2016. The primary aim was to evaluate the effects of these interventions on interlimb volume difference [29]. This report is a secondary analysis examining the effects of the trial on insulin-related biomarkers of breast cancer recurrence.

### 2.3. Exercise Intervention

All exercise sessions could be performed within the participant’s home and weight-adjustable dumbbells were provided for the weight training component of the exercise intervention. Participants were asked to engage in two weight training sessions and 180 min of aerobic exercise per week. For the first six weeks, participants received in-person, on-site weekly instruction from certified fitness professionals that focused on the proper and safe execution and gradual increase of the prescribed resistance exercises. From weeks 7 to 52, participants received monthly in-person sessions at the study site, in addition to performing two weight training and 6 aerobic exercise sessions per week at home. Resistance for weight training exercises was gradually increased throughout the intervention. Aerobic exercise prescription remained constant the entire intervention. Behavioral counseling was included in the monthly sessions with the goal of maximizing adherence. All participants were asked to keep a log of their exercises performed. Exercise trainers called participants weekly to provide behavioral counseling, answer questions, and check on adherence.

### 2.4. Weight Loss Intervention

The first 24 weeks of the intervention included weekly group meetings and meals provided using NutriSystem^®^ to promote weight loss. Meetings were designed to increase adherence through behavioral modification lessons with a different topic each week such as goal setting, problem solving, etc. During the first 20 weeks, daily caloric intake was restricted to 1200–1500 kcal/day through the provision of Nutrisystem^®^ shelf stable meals and snacks. From weeks 20–24 participants were encouraged to transition to purchasing their own food from the grocery store while maintaining 1200–1500 kcal/day. During the following 28 weeks, participants increased their caloric intake to 1700–2000 kcal/day with the goal of maintaining the weight they had lost in the initial 24 weeks. In this 28-week period there were monthly group meetings and weekly individual calls with a registered dietitian. 

### 2.5. Exercise and Weight Loss Intervention

Participants in this group engaged in the same exercise protocol as the exercise intervention group. After week 6, they continued the exercise protocol in addition to starting the weight loss program. Detailed study design has been published previously [28].

### 2.6. Control Group

Participants in the control group were directed to the American Cancer Society website for all diet-related questions. For exercise-related questions they were referred to their physician. Participants were asked to continue with the exercise regimen they had prior to enrollment in the study. However, all participants were sedentary (self-report) at study entry.

### 2.7. Biomarker Assays

Trained medical staff collected 12-h fasting blood samples at the baseline and 12-month follow-up clinic visit. EDTA plasma samples, aliquoted and stored at −80 °C until assay preparation were used for biomarker measures. Laboratory personnel were blinded to participants’ study groups. Plasma samples were prepared and analyzed according to standard methods and quality control procedures. Fasting plasma glucose concentrations were measured using a glucose oxidase method (YSI 2900 Biochemistry Analyzer). Plasma insulin and C-peptide concentrations were determined using high-sensitivity immunoassays (Meso Scale Discovery, catalog #K15164C and # K151X5D, respectively). Intra-plate and inter-plate coefficients of variance (CV), respectively, were: insulin (3.5%, 6.5%), glucose (2.1%, 3.2%), and C-peptide (4.1%, 9.6%).

### 2.8. Measurements

Body weight was measured, for analytic purposes, at baseline and 12 months. Height was measured at baseline. Dual energy x-ray absorptiometry (DEXA) and blood draws were performed at baseline and 12 months. The treadmill exercise test was conducted according to the modified Bruce protocol [31]. Homeostatic model assessment (HOMA2) insulin resistance (IR) and beta-cell function (β) were calculated using the HOMA2 calculator released by the Diabetes Trials Unit, University of Oxford [32]. Insulin (pmol//L) and glucose (mmol/L) values were used to calculate HOMA2-IR and C-peptide values (nmol/L) and glucose (mmol/L) were used to calculate HOMA2-β.

### 2.9. Statistical Analysis

Two hundred and six of the 351 women who participated in the WISER Survivor trial were included in this analysis. Participants were excluded if baseline or follow-up biomarker data was not available (*n* = 127) (due to inadequate sample availability to assay, loss to follow up, or not meeting quality control), or self-reported as non-fasting prior to blood draw (*n* = 10), or if insulin, C-peptide or glucose fell outside of the usable range for the HOMA2 formulas (indicating a non-fasting blood draw, *n* = 8) (Figure 1). Of the 206 participants included, 199 completed baseline and follow-up DEXA scans, and 186 completed baseline and 12-month treadmill testing. The Shapiro–Wilk test was used to test for normality and logarithmic transformations were performed accordingly. The Kruskal–Wallis and Chi-Squared test were used to test for differences between groups at baseline. One-way ANOVA with post hoc Bonferroni-adjusted differences were compared between biomarkers at baseline. A multiple linear regression model was used to examine main effects of the intervention using log-transformed baseline biomarker of interest and use of glucose-related medication as covariates. A multiple linear regression model was used to examine differences between tertiles using log-transformed baseline biomarker of interest, age, intervention arm, glucose-related medication use, and race as covariates. A logistic regression model was used to determine odds ratios for impaired baseline C-peptide and glucose levels returning to normal at 12 months). Clinically impaired C-peptide values were defined as falling outside 0.78–1.89 ng/mL [33]. Clinically impaired fasting glucose levels were defined as ≥100 mg/dL [34]. The odds ratio represents the probability of returning to a normal C-peptide or glucose range at 12-months compared to the control group. Baseline biomarker of interest, age, change in fat mass, change in lean body mass and change in treadmill time were used as covariates. The datasets generated during and/or analyzed during the current study are available from the corresponding author on reasonable request.

## 3. Results

Participant characteristics are summarized in Table 1. The study population was 33% Black or other minority, on average with a BMI indicating overweight or obesity, greater than 5 years since diagnosis, and college educated. Forty-four percent of participants were on endocrine therapy and 8% were taking medication related to glucose management during the study. Adherence to aerobic exercise was reported at 145 ± 72 and 163 ± 94 min per week in the exercise group and combined group, respectively. Attendance at supervised exercise sessions was 85.8 ± 18% and 88.9 ± 29% in the exercise group and combined group, respectively.

### 3.1. Intervention Effects

Table 2 displays the baseline and follow up values of plasma biomarkers. There were no differences in levels between groups at baseline. At follow up, the weight loss and combined intervention groups experienced a significant decrease in insulin, C-peptide, HOMA2-IR, and HOMA2-β compared to the control group. Insulin decreased by 22.3% in the weight loss group and by 18.5% in the combined group. C-peptide decreased by 16.7% in the weight loss group, and by 13% in the combined group. HOMA2-IR decreased by 20% in the weight loss group, and by 16.7% in the combined group. HOMA2-β decreased by 3.2% in the exercise group, 2.2% in the weight loss group, and 5.6% in the combined group. The exercise group experienced a 4.5% increase in glucose.

#### 3.1.1. Effects of Change in Body Composition and Treadmill Endurance

Table 3 displays the change in biomarker by categories of weight loss, and by tertiles of change in fat mass, lean mass, time on treadmill (i.e., fitness). Compared to those who lost between 0–5% of their weight, participants who lost ≥10% of their baseline weight experienced significant changes in insulin, C-peptide, and HOMA2-IR. Similarly, in tertiles 2 and 3 of change in fat mass (≥1.3 kg fat loss) had a significant change in insulin, C-peptide, and HOMA2-IR. Mean levels of lean mass change were 0.1 ± 1.8 kg, 0.41 ± 2.5 kg, −0.83 ± 3 kg, −1.2 ± 2.5 kg in the control, exercise, weight loss and combined intervention groups, respectively. The addition of the exercise intervention to caloric restriction did not mitigate the loss of lean mass. Participants in the upper tertile of change in lean mass (≥0.7 kg lean mass gained) experienced significantly less improvement in insulin, C-peptide, glucose, and HOMA2-IR, compared to participants that lost lean mass. Participants whom improved their treadmill test duration, “fitness capacity”, by at least 31 s experienced significant decreases in insulin and HOMA-IR.

#### 3.1.2. Normalization of C-Peptide

We observed that only the combined intervention group significantly increased odds of improving C-peptide levels from clinically impaired, to normal (Table 4). The odds (95% CI) of returning to normal C-peptide range for the exercise was 2.3 (0.8–6.6), 2.9 (1.0–9.0) in the weight loss group and 4.5 (1.4–14.1) for the combined group. Incidence of impaired C-peptide at follow up for participants who had normal baseline values was 40% for control, 25% for the exercise group, 12.5% for the weight loss group and 13.6% in the combined group. A regression to the mean analysis for impaired fasting glucose (>100 mg/dL) at baseline demonstrated that none of the intervention groups significantly increased the odds of returning to a healthy fasting glucose (<100 mg/dL) at 12 months.

## 4. Discussion

We observed that, compared to the control condition, weight loss with or without exercise led to significant reductions in insulin and insulin resistance. Insulin and associated metabolic pathways are associated with breast cancer recurrence and hypothesized to be a mechanistic driver of cancer. Obesity and lack of physical activity are common modifiable risk factors amongst breast cancer survivors and are strongly associated with hyperinsulinemia and insulin resistance. Using lifestyle modification in the form of exercise and weight loss, survivors can reduce insulin and insulin resistance through altered body composition. With an increasing number of breast cancer survivors, an increased emphasis on lifestyle modification to reduce recurrence and the sequela of breast cancer and breast cancer treatment is warranted. The results of this study demonstrate that reduction of insulin levels and increased insulin sensitivity is more effectively accomplished with a weight loss intervention than an exercise intervention when compared to the control. However, for participants with clinically impaired C-peptide levels, a combination of exercise and weight loss may be necessary to normalize C-peptide levels after taking into account changes in body composition and age.

We observed decreased insulin, C-peptide, HOMA2-IR, and HOMA2-β levels with a weight loss intervention and a combined weight loss and exercise intervention. Previous studies utilizing combined exercise and weight loss interventions in breast cancer survivors have observed similar results [26,27]. Unlike prior studies, this study isolates the individual and combined effects of exercise and weight loss to demonstrate that weight loss alone, or in combination with exercise, is more effective at improving biomarkers of insulin levels and insulin sensitivity than an exercise-only intervention.

The exercise prescription in the WISER Survivor trial was prescribed with specificity for lymphedema outcomes. We did not observe any change in insulin levels or insulin resistance in the exercise group. Yet, change has been observed in other trials using supervised aerobic and strength training interventions [25,35]. Thus, it is possible the exercise prescription was not appropriate for outcomes related to insulin levels and insulin resistance or the lack of supervision during the aerobic exercise portion may have led to bias in reporting the amount of aerobic exercise completed. Additionally, given that the upper tertile of fitness capacity in this study was set at an increase of 31 s on the Modified Bruce Treadmill protocol, it is likely that while participants may have increased their step count and physical activity minutes, they did not increase their exercise intensity and thus fitness capacity. Indeed, we observed that participants in the upper tertile for increased fitness capacity significantly decreased both fasting insulin and HOMA2-IR. Although not significant, C-peptide, fasting glucose and HOMA2-B did trend towards significant with increasing fitness capacity. This suggests that these biomarkers can be significantly decreased with an exercise intervention prescribed to increase fitness capacity via increased exercise intensity, or, a supervised exercise program. However, we cannot ignore that weight loss and increased fitness capacity are coupled and more work needs to be done to assess the ability of exercise, independent of weight loss, to improve insulin sensitivity in this patient population.

Independent of intervention group, we observed that a 10% or greater weight loss improves biomarkers of insulin sensitivity and insulin resistance. Our results align with Fabian et al. and their 6-month combined exercise and weight loss intervention that demonstrated weight loss of >10% resulted in improvements in serum and breast tissue biomarkers, including insulin levels, compared to <10% weight loss [36]. We demonstrate that exercise may not be necessary to achieve this 10% weight loss or significantly improve insulin levels or insulin sensitivity. Further assessment of body composition indicates that weight loss specific to fat mass (>1.3 kg) was sufficient to improve levels of insulin, C-peptide, and HOMA2-IR. However, 5.0 kg or more of fat mass loss was required for a significant decrease in HOMA2-β. The decrease in HOMA2-β indicates a decrease in beta cell function, however, it is likely that beta cell function is not decreased, rather, the decrease in insulin resistance is driving HOMA2-β down. Decreased adiposity lowers insulin resistance. This means that the beta cells of the pancreas are simply producing less insulin because the lowered insulin resistance has decreased the demand for insulin production.

Lifestyle interventions that improve body habitus alter biomarkers of insulin and insulin resistance, which are mechanistic contributors to breast cancer pathogenesis [37]. This can explain why women who engage in these interventions have better outcomes with respect to breast cancer survival [20,21], adverse sequelae of cancer/cancer treatment [19], and recurrence [18]. This study reinforces the current association between markers of insulin resistance and body composition.

When implementing lifestyle interventions, it is important to establish behavior change goals that are both attainable and have significant health benefits. Our results suggest that while any percentage of weight loss improves markers of insulin resistance, attaining a >10% weight loss is key. Unfortunately, weight loss of this size is at the upper range of what can be expected for most individuals treated with lifestyle modification or an anti-obesity medication. Additionally, the improvement in markers of insulin resistance in this patient population is tightly coupled with the magnitude of change in body composition. The most effective way to alter body composition and improve insulin resistance is through a combined intervention of caloric restriction and exercise. Indeed, for participants with impaired C-peptide levels, the combination intervention was the only intervention arm to return a significant number of participants to normal C-peptide levels. Fasting glucose levels did not respond to the interventions in the same way as insulin, C-peptide and HOMA-IR. Unlike the exercise, weight loss and combined interventions by Mason et al., our interventions were ineffective at returning impaired fasting glucose levels to normal levels [38]. The differences between the Mason et al. trial and the WISER Survivor study are that the Mason et al. trial diet intervention had a target weight loss of 10% and the exercise intervention had more supervision compared to the WISER Survivor trial exercise program. These differences may account for the lack of improvement in fasting glucose levels in our study population.

A strength of the WISER Survivor Trial was the diversity of the cohort. We were successful in recruiting a cohort with the largest number of Black breast cancer survivors reported to date [30]. The study sample is highly representative of the general US population at present. Additionally, the use of the 2 × 2 factorial design to examine relative effects of exercise and weight loss, or their combination on biomarkers of breast cancer recurrence (insulin, C-peptide, glucose, and clinically relevant indices such as HOMA2-IR and HOMA2-β) was novel. However, the WISER Survivor Trial was designed for a primary outcome associated with lymphedema. The exercise prescription was tailored specifically for slow progressive resistance training and a recommendation for 180 min per week of aerobic exercise training. The specificity of the WISER Survivor Trial exercise prescription is therefore a limitation for assessment of changes in insulin biomarkers. Yet, for many breast cancer survivors, lymphedema and insulin resistance may be co-occurring. Thus, the collective observations from the WISER Survivor Trial are important for both breast cancer survivors, clinicians, and exercise physiologists.

## Figures and Tables

**Figure 1 nutrients-13-03108-f001:**
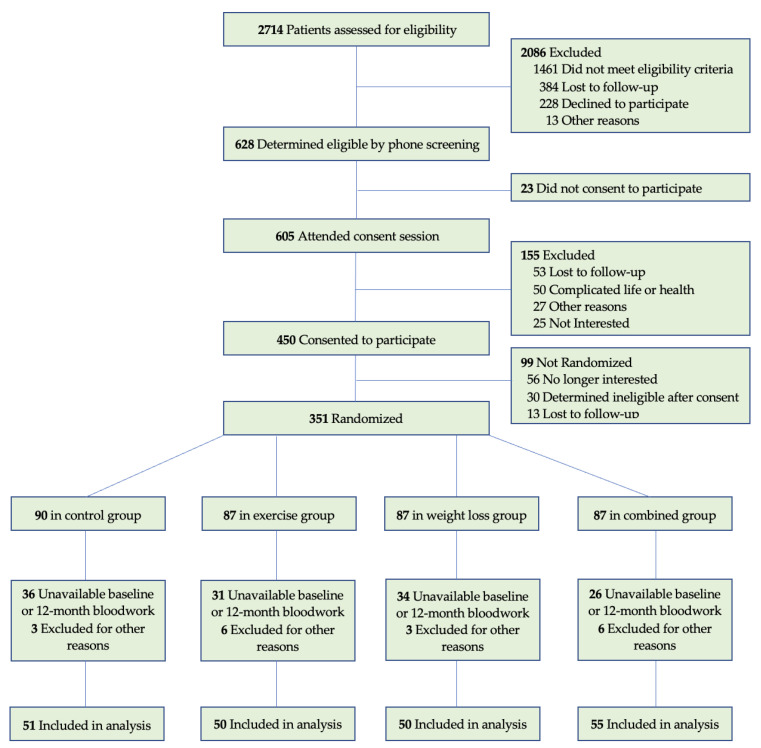
Two hundred and six of the 351 women who participated in the WISER Survivor trial were included in this analysis. Participants were excluded if baseline or follow-up biomarker data was not available (*n* = 127) (due to inadequate sample availability to assay, loss to follow up, or not meeting quality control), or self-reported as non-fasting prior to blood draw (*n* = 10), or if insulin, C-peptide or glucose fell outside of the usable range for the homeostatic model assessment 2 (HOMA2) formulas (indicating a non-fasting blood draw, *n* = 8).

**Table 1 nutrients-13-03108-t001:** Demographic and clinical characteristics of the groups at baseline.

	Cohort *n* = 206	Control *n* = 51	Exercise *n* = 50	Weight Loss *n* = 50	Combined *n* = 55	*p* Value
Age, y	59.9 ± 8.9	60.5 ± 8.9	59.3 ± 8.8	59.7 ± 8.8	60.2 ± 9.2	0.90
BMI, kg/m^2^	33.8 ± 5.9	33.3 ± 5.3	34.5 ± 7.0	34.1 ± 5.7	33.4 ± 5.8	0.86
Time since diagnosis, y	7.8 ± 5.3	8.8 ± 5.4	7.4 ± 5.3	7.6 ± 5.6	7.4 ± 4.9	0.41
Race						
Non-Hispanic White	138 (67)	37 (72.5)	33 (66.0)	34 (68.0)	34 (61.8)	
Black	61 (29.6)	12 (23.5)	16 (32.0)	16 (32.0)	17 (30.9)	
Other	7 (3.4)	2 (3.9)	1 (2.0)	0	4 (7.3)	0.45
Education						
High school diploma	36 (17.5)	11 (21.6)	7 (14.0)	5 (10.0)	13 (23.6)	
Some college	67 (32.5)	16 (31.4)	14 (28.0)	21 (42.0)	16 (29.1)	
College education	103 (50)	24 (47.1)	29 (58.0)	24 (48.0)	26 (47.3)	0.38
Endocrine therapy						
Aromatase inhibitors	70 (33)	17 (33.3)	16 (32.0)	19 (38.0)	18 (32.7)	0.92
Tamoxifen	20 (9.7)	4 (7.8)	8 (16.0)	5 (10.0)	3 (5.4)	0.31
Anti-Diabetes Medication	17 (8.2)	1 (2.0)	4 (8.0)	7 (14.0)	5 (9.1)	0.18

Data presented as mean ± SD or *n* (%).

**Table 2 nutrients-13-03108-t002:** Main intervention effects on biomarkers of insulin sensitivity.

	Control	Exercise	Weight Loss	Combined
Insulin (uIU/mL)				
Baseline	17.8 ± 11.5	16.5 ± 10.2	17.5 ± 8.3	16.2 ± 11.4
12-Months	18.1 ± 10.6	16.8 ± 9.9	13.5 ± 7.3 ^a^	13.2 ± 9.6 ^a^
Change	0.3	0.3	−4.0	−3.0
C-peptide (ng/mL)				
Baseline	2.4 ± 1.0	2.3 ±1.0	2.4 ± 1.0	2.3 ± 1.2
12-Months	2.5 ± 0.9	2.5 ± 1.1	2.0 ± 0.9 ^a^	2.0 ± 1.1 ^a^
Change	0.1	0.2	−0.4	−0.3
Glucose (mg/dL)				
Baseline	103 ± 14.5	111 ± 31.7	110 ±14.9	102 ± 12.7
12-Months	99.7 ± 15.1	116 ± 40.6 ^a^	106 ±17.5	101 ± 14.0
Change	−3.3	5.0	−4.0	−1.0
HOMA2-IR				
Baseline	2.0 ± 1.3	1.9 ± 1.2	2.0 ± 0.95	1.8 ± 1.3
12-Months	2.0 ± 1.2	2.0 ± 1.2	1.6 ± 0.8 ^a^	1.5 ± 1.1 ^a^
Change	0	0.1	−0.4	−0.3
HOMA2-β				
Baseline	110 ± 35.4	103 ± 33.8	97.2 ± 31.8	107 ± 38.1
12-Months	121 ± 38.1	99.7 ± 34.4 ^a^	95.1 ± 32.6 ^a^	101 ± 39.5 ^a^
Change	11	−3.3	−2.1	−6.0

Data presented as mean ± SD. ^a^
*p* < 0.05. Homeostatic model assessment 2 (HOMA2) insulin resistance (IR), and HOMA2 beta-cell function (β).

**Table 3 nutrients-13-03108-t003:** Changes in biomarkers of insulin sensitivity among WISER Survivor participants stratified according to weight loss and tertiles of change in fat mass, change in lean mass, and change in aerobic fitness.

	Weight Loss (%) *		
	<0–5 (Reference)	≥5–10	≥10
*n*	70	30	47
Δ Insulin (uIU/mL)	−2.79 (8.5)	−3.16 (6.7)	−5.03 (7.7) ^a^
Δ C-peptide (ng/mL)	−0.08 (0.7)	−0.17 (0.6)	−0.55 (0.7) ^a^
Δ Glucose (mg/dL)	−0.20 (26.1)	0.07 (12.8)	−8.08 (11.6)
Δ HOMA2-IR	−0.31 (0.9)	−0.36 (0.8)	−0.60 (0.9) ^a^
Δ HOMA2-β	1.20 (35.8)	−4.69 (23.4)	−4.35 (21.9)
	**Δ Fat Mass (kg) ****		
Tertile	1 (Reference)	2	3
Range (kg)	+15.5 to −1.2	−1.3 to −4.9	−5.0 to −30
Mean, [Median (SD)]	1.4, [0.95 (2.6)]	−2.7, [−2.6 (0.97)]	−9.1, [−7.7 (4.3)]
*n*	66	66	67
Δ Insulin (uIU/mL)	2.78 (7.9)	−2.39 (8.1) ^a^	−5.52 (7.6) ^a^
Δ C-peptide (ng/mL)	0.32 (0.7)	−0.08 (0.73) ^a^	−0.53 (0.66) ^a^
Δ Glucose (mg/dL)	3.82 (27.1)	0.15 (13.4)	−5.87 (12.3)
Δ HOMA2-IR	0.33 (0.91)	−0.26 (0.89) ^a^	−0.65 (0.86) ^a^
Δ HOMA2-β	5.45 (27.6)	−0.43 (35.1)	−6.33 (23.4) ^a^
	**Δ Lean Body Mass (kg) ****		
Tertile	1 (Reference)	2	3
Range (kg)	−7.7 to −1.298	−1.297 to 0.68	0.7 to 13.3
Mean, [Median (SD)]	−3.1, [−2.8 (1.4)]	−0.4, [−0.4 (0.5)]	2.3, [1.8 (1.8)]
*n*	67	66	66
Δ Insulin (uIU/mL)	−3.18 (7.6)	−1.6 (9.1)	−0.38 (8.8) ^a^
Δ C-peptide (ng/mL)	−0.26 (0.8)	−0.08 (0.8)	0.04 (0.7) ^a^
Δ Glucose (mg/dL)	−6.95 (19.7)	2.16 (20.9) ^a^	2.9 (15.1) ^a^
Δ HOMA2-IR	−0.38 (0.9)	−0.17 (1.0)	−0.03 (1.0) ^a^
Δ HOMA2-β	−0.26 (25.2)	−2.08 (25.3)	0.93 (36.6)
	**Δ Time on Treadmill (s) *****		
Tertile	1 (Reference)	2	3
Range (sec)	−711 to −65	−64 to 28	31 to 752
Mean, [Median (SD)]	−194, [−166 (133)]	−13.4, [−0.5 (27.8)]	163, [117 (151)]
*n*	62	62	62
Δ Insulin (uIU/mL)	0.11 (8.6)	−1.67 (9.5)	−4.17 (7.3) ^a^
Δ C-peptide (ng/mL)	−0.03 (0.7)	−0.07 (0.9)	−0.26 (0.6)
Δ Glucose (mg/dL)	−0.43 (14.7)	−0.76 (27.0)	−1.57 (13.4)
Δ HOMA2-IR	0.01 (1.0)	−0.19 (1.0)	−0.48 (0.1) ^a^
Δ HOMA2-β	2.54 (26.7)	0.20 (34.9)	−4.65 (22.1)

Data presented as mean ± SD. ^a^
*p* < 0.05. * Participants (*n* = 59) who gained weight between baseline and 12-months were excluded. ** Participants (*n* = 7) without baseline and 12-month DEXA measurements were excluded. *** Participants (*n* = 20) without baseline and 12-month treadmill testing were excluded. (*n* = 20).

**Table 4 nutrients-13-03108-t004:** Odds ratio (95% CI) for C-peptide and glucose changing from clinically abnormal at baseline to normal at 12-month follow-up. Normal C-peptide and glucose range were 0.78–1.89 ng/mL and <100 mg/dL, respectively.

	Adjusted Model C-Peptide	Adjusted Model Glucose
Control	1.0 *n* = 33 ^b^, 37 ^c^	1.0 *n* = 28 ^b^, 25 ^c^
Exercise	2.3 (0.8–6.6) *n* = 32 ^b^, 29 ^c^	0.44 (0.2–1.2) *n* = 27 ^b^, 32 ^c^
Diet	2.9 (1.0–9.0) *n* = 32 ^b^, 23 ^c^	0.86 (0.3–2.4) *n* = 39 ^b^, 31 ^c^
Combined	4.5 (1.4–14.1) ^a^ *n* = 31 ^b^, 23 ^c^	0.75 (0.3–2.2) *n* = 29 ^b^, 28 ^c^

Data presented as odds ratio (95%, CI). ^a^
*p* < 0.05. ^b^ = number of participants abnormal at baseline. ^c^ = number of participants abnormal at 12-month.

## Data Availability

All data used in this study can be made publicly available upon reasonable request.

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
