# Peer review of "WISER Survivor Trial: Combined Effect of Exercise and Weight Loss Interventions on Insulin and Insulin Resistance in Breast Cancer Survivors"

_nutrients, 2021, doi:10.3390/nu13093108_

Round 1
Reviewer 1 Report
The paper is clear and well written
I only have some minor remarks:
Page 2, lines 79-80: “This report is a secondary analysis examining the effects of the trial on biomarkers of breast cancer recurrence”. Please explain
Discussion
Lines 280-285: “Lifestyle intervention in the form of exercise and weight loss have utility to improve outcomes with respect to breast cancer survival, adverse sequelae of cancer/cancertreatment, and recurrence. Insulin and the insulin pathways are mechanistic contributors in breast cancer pathogenesis, which supports why women who exercise and maintain a healthy body habitus have improved outcomes. This study reinforces the current association between markers of insulin resistance and body composition”
These concepts should be better analyzed and appropriately referenced.
Reviewer 2 Report
Thank you very much for that interesting manuscript. I have have just two comments.
A flow chart of the study would be useful to the reader.
And did the authors also analysed breast cancer biomarker and checked the changes? Would be for the reader
